# Tunable Soft Prompts are Messengers in Federated Learning

**Chenhe Dong**[1,*,†]   **Yuexiang Xie**[2,*]   **Bolin Ding**[2]   **Ying Shen**[1,‡]   **Yaliang Li**[2,‡]
[1]Sun Yat-sen University   [2]Alibaba Group
dongchh@mail2.sysu.edu.cn   {yuexiang.xyx, bolin.ding, yaliang.li}@alibaba-inc.com
sheny76@mail.sysu.edu.cn

## Abstract

Federated learning (FL) enables multiple participants to collaboratively train machine learning models using decentralized data sources, alleviating privacy concerns that arise from directly sharing local data. However, the lack of model privacy protection in FL becomes an unneglectable challenge, especially when people want to federally finetune models based on a proprietary large language model. In this study, we propose a novel FL training approach that accomplishes information exchange among participants via tunable soft prompts. These soft prompts, updated and transmitted between the server and clients, assume the role of the global model parameters and serve as messengers to deliver useful knowledge from the local data and global model. As the global model itself is not required to be shared and the local training is conducted based on an auxiliary model with fewer parameters than the global model, the proposed approach provides protection for the global model while reducing communication and computation costs in FL. Extensive experiments show the effectiveness of the proposed approach compared to several baselines. We have released the source code at https://github.com/alibaba/FederatedScope/tree/fedsp/federatedscope/nlp/fedsp.

## 1 Introduction

Large language models (LLMs) (Radford et al., 2019; Brown et al., 2020; Zhang et al., 2022; Chowdhery et al., 2022; Scao et al., 2022; Zeng et al., 2022; OpenAI, 2023; Touvron et al., 2023) have been witnessed incredible progress in the recent years, which is tied to the support of large amounts of training corpus and computation resources. To further promote the development of LLMs and broaden their applications, how to make good use of private and decentralized data can be one of the critical steps, which brings both opportunities and challenges to federated learning (FL) (Konečný et al., 2016; McMahan et al., 2017; Yang et al., 2019; Kairouz et al., 2021).

The main idea of FL is that multiple participants collectively train a global model using their local data, and the updates generated by each participant's local training process are then aggregated together to optimize the global model. On the one hand, FL offers a feasible solution for finetuning LLMs by utilizing multiple data sources without private data leakage. Participants are allowed to conduct local training processes based on their private data, and then share the learned knowledge through the exchange of model updates. In this way, privacy concerns, raised by directly sharing private data, can be alleviated.

However, on the other hand, the application of FL can be constrained by concerns regarding model privacy, especially when the finetuning process relies on proprietary LLMs. In particular, during each training round, the up-to-date global model would be distributed to participants, which goes against the interests of model owners and might be deemed unacceptable in real-world scenarios.

Such conflicts between protecting data privacy and model privacy are attributed to the training paradigm of FL, which involves sharing model parameters to accomplish the knowledge exchange process for collaboratively training models. To address this challenge, we propose FEDSP, a novel FL training approach that leverages *tunable soft prompts* to enable the exchange of useful knowledge among participants, achieving both data privacy and model privacy protection at the same time.

Soft prompts can be regarded as some additional tunable parameters plugged into the model, which are updated to capture knowledge from downstream tasks while keeping the model parameters frozen in *parameter-efficient finetuning* (PEFT) al-

---

*Equal contribution.
†Work done at Alibaba.
‡Corresponding authors.

gorithms (Houlsby et al., 2019; Lester et al., 2021; Li and Liang, 2021; Hu et al., 2021; Zaken et al., 2021). We incorporate these tunable soft prompts in FL, serving as messengers among participants to deliver knowledge learned from local data and contained in the global model.

Specifically, at the start of an FL course, a server broadcasts an auxiliary model (which typically has much fewer parameters than the global model) to participating clients. Then in each training round, the server sends up-to-date tunable soft prompts to selected clients, and these clients combine the received soft prompts with their maintained auxiliary models. After that, each client performs local training to accomplish the process consisting of (i) *Global Model Alignment*, in which clients update the auxiliary models to align them with the representative capabilities of the global model, using the up-to-date soft prompts; and (ii) *Local Knowledge Capturing*, in which clients freeze their auxiliary models and finetune the soft prompts to capture useful knowledge from their local data. These updated soft prompts are sent back to the server once the local training is complete. The server is responsible for aggregating and optimizing these soft prompts to drive the next training round.

In contrast to the existing FL training paradigm that entails sharing the parameters of the global model, the proposed FEDSP suggests exchanging tunable soft prompts during training rounds, which ensures privacy protection for the global model. Meanwhile, clients are only expected to update the auxiliary models and soft prompts in the local training process, which mitigates both computation and communication overhead compared to federally finetuning of large global models like LLMs.

We conduct a series of experiments with two LLMs (GPT2-XL and OPT-1.3B) on seven benchmarking datasets, showing that FEDSP achieves competitive performance compared to baseline methods while significantly reducing the model size by $14.5\times/8.5\times$ on GPT2-XL/OPT-1.3B. These experimental results demonstrate the effectiveness and advantages of the proposed idea that using tunable soft prompts as knowledge messengers in FL.

## 2   Preliminary

Because of the huge model size (billions of parameters), large language models (LLMs) (Chowdhery et al., 2022; Zeng et al., 2022; OpenAI, 2023; Touvron et al., 2023; Zhao et al., 2023; Chen et al.,

2023) are usually kept at the cloud server who owns adequate resources for inferring, finetuning, and training. Nowadays, users of LLMs have to send the instructions to the cloud server and wait for the model-generated results, or upload their private data[1] for finetuning the LLMs on their downstream tasks. The server can also benefit from the user instructions and human feedbacks to continually improve the model performance.

Such usage of LLMs might bring privacy issues from two different perspectives. For the users of LLMs, they might not be allowed to upload their private data to the server, especially in some scenarios with highly sensitive information such as health care, so-called *data privacy*. For the cloud servers (i.e., the model owners), they tend to keep the proprietary language model private to ensure their interests, so-called *model privacy*. As a result, how to provide protection for both data privacy and model privacy is a practical and challenging problem when training, finetuning, and deploying LLMs in real-world applications.

Federated Learning (FL) (Konečný et al., 2016; McMahan et al., 2017) is one of the considered solutions for alleviating the aforementioned data privacy issue. Different from the centralized training paradigm that needs to gather users' local data for training a global model, FL proposes to aggregate participants' model updates to avoid directly sharing the private data. To be more specific, at each training round, a server broadcasts the up-to-date global model to the participating clients. Each client updates the received global model based on its local data, and then sends the model updates back to the server for performing federated aggregation. Formally, the federated aggregation at the $t$-th training round can be defined as:

$$\boldsymbol{w}_g^t = \sum_{k=1}^{K} \frac{N_k}{N} \boldsymbol{w}_k^t, \qquad (1)$$

where $K$ is the number of clients, $N$ is the total amount of training data, $\boldsymbol{w}_k$ is the updated model parameters of the $k$-th client, and $\boldsymbol{w}_g$ is the aggregated global model parameters.

Nevertheless, as the global model is required to be shared in the training process of FL, the model privacy issue has not been well addressed yet, which motivates us to make an improvement in providing both data privacy protection and model privacy protection in FL.

---

[1]https://platform.openai.com/docs/guides/fine-tuning

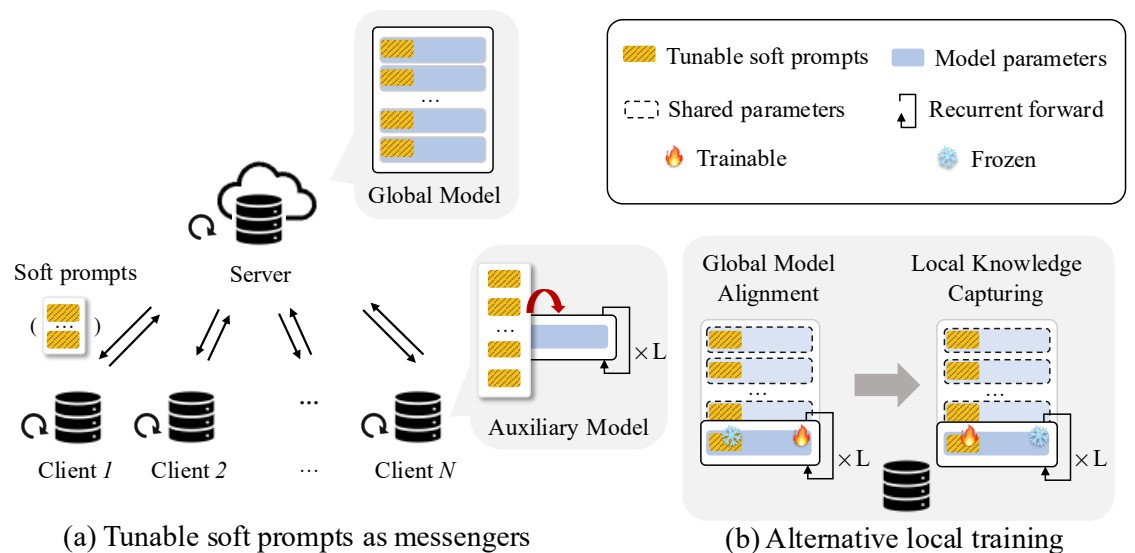

Figure 1: Overall architecture of the proposed FEDSP.

## 3 Methodology

In this section, we describe the details of the proposed FEDSP. The intuition behind FEDSP is to adopt some tunable soft prompts to replace the shared global model in the training process of FL, which serves as messengers to deliver useful knowledge in the local data and the global model to the server and clients, as introduced in Section 3.1. With the help of these tunable soft prompts, clients are expected to perform local training for updating an auxiliary model and the soft prompts alternatively, and send the updated soft prompts to the servers for sharing the knowledge learned from their local data (more details in Section 3.2). The overall architecture of the proposed FEDSP is illustrated in Figure 1.

### 3.1 Tunable Soft Prompts

The existing FL algorithms (Yang et al., 2019; Kairouz et al., 2021) leverage the global model parameters to exchange the useful knowledge learned from participants' local data. However, it might raise the model privacy issue when the training process is conducted on a private model, such as finetuning a proprietary large language model on downstream tasks. Inspired by the parameter-efficient finetuning (PEFT) algorithms (Lester et al., 2021; Li and Liang, 2021), we propose to adopt tunable soft prompts to tackle such a model privacy issue.

These soft prompts are continuous vectors added to the LLMs, serving as instructive contexts to influence the generation process of LLMs by steering the probability distribution of the next token. By optimizing these soft prompts over the training datasets via backward propagation, the knowledge contained in the data can be captured and condensed into their parameters.

In this study, we follow the tunable soft prompts proposed by PREFIX-TUNING (Li and Liang, 2021). The tunable soft prompts would be updated and transmitted between the server and clients for exchanging useful knowledge. Note that soft prompts should be plugged into a model first, and are updated based on the training data while other parameters of this model are kept frozen. For the server that owns the global model, it is straightforward and reasonable to add tunable soft prompts to the global model. However, adding and maintaining soft prompts for the clients becomes challenging, since clients could not access the global model due to the model privacy issue.

In order to tackle such a challenge, each client is expected to integrate the tunable soft prompts with an auxiliary model and perform an alternative local training process. Before we introduce more details of the local training, we first pay more attention to the auxiliary model. In the proposed FEDSP, the auxiliary model should be tied to the global model but could not cause model privacy leakage, and had better be a shadow model that has fewer parameters than the global model considering data quantity and computation resources of a client can be limited in some application scenarios.

Motivated by the aforementioned requirements, we construct the auxiliary model by substantially

reducing the depth of the global model and applying the cross-layer parameter sharing, inspired by the main idea of Albert (Lan et al., 2020). Given that most of the LLMs adopt the transformer-based architecture (Vaswani et al., 2017), such a design of the auxiliary model is suitable for plugging and tuning the soft prompts shared by the server.

The example illustrated in Figure 1(a) shows that, the server owns a global model with total $L$ layers and each layer is concatenated with some prefix prompts, while each client keeps an auxiliary model with only 1 layer and plugged with $L$-layer prefix prompts. Using the auxiliary model, clients only need to update the tunable soft prompts based on their local data and exchange the updated soft prompts with others to accomplish the FL course. Thus the proposed FEDSP provides privacy protection for the global model as the global model is only maintained by the server and won't be shared. Meanwhile, clients need fewer computation and communication resources for updating the auxiliary model and soft prompts in the local training process compared to those of training the global model as in previous studies.

In the rest of this section, we describe the local training process of clients in the proposed FEDSP, showing how to deliver useful knowledge via updating and exchanging the tunable soft prompts.

## 3.2 Training Procedure

At the beginning of an FL course, the server first initializes the auxiliary model according to its global model, and broadcasts the generated auxiliary model to all the participating clients. Then the server sends the up-to-date soft prompts to the selected clients at each training round, while the clients finetune the received soft prompts and also update the auxiliary model accordingly. The updated soft prompts are uploaded to the server for performing federated aggregation, and then the aggregated soft prompts are plugged into the global model and further optimized by the server. Finally, the up-to-date soft prompts are sent to clients to start a new training round.

Note that compared to the existing FL training paradigms that exchange the global model, in the proposed FEDSP, the models kept at the server (i.e., the global model) and clients (i.e., the auxiliary models) have different model sizes but are plugged with the same prompts. As a result, to narrow the optimization gap and make the federal training

process meaningful, we propose to provide a good initialization of the auxiliary model and perform an alternative local training process for the clients.

**Initialization** The different model sizes between the server and clients might lead to the misaligned representation, and further cause the updating of the soft prompts mismatch between the server and clients. Such a mismatch seriously hinders effective knowledge exchange between the server and clients in an FL course.

To alleviate such a mismatch, the server leverages knowledge distillation (KD) (Hinton et al., 2015) techniques to initialize the auxiliary models before sharing them with clients. The main idea is to align the representations of the student models (i.e., auxiliary models) with the large teacher model (i.e., the global model), which is widely used in previous studies (Cheng et al., 2021; Xiao et al., 2023) for representation alignment. Formally, given the last hidden states of the teacher model $\mathbf{H}^T$ and the student model $\mathbf{H}^S$, the loss function of KD can be defined as:

$$\mathcal{L} = \text{MSE}(\mathbf{H}^T, \mathbf{W}^S \mathbf{H}^S), \tag{2}$$

where MSE is the *mean square error* function, $\mathbf{W}^S$ is a learnable transformation matrix.

Notably, the tunable soft prompts are not added to the global model and auxiliary models during the KD process. And the aforementioned KD process is only performed by the server at the beginning of an FL course, whose computation cost is affordable. After initialization, the auxiliary models are broadcast to all participating clients and updated by clients according to their local data independently.

**Alternative Local Training** When receiving the auxiliary models, which have fewer layer numbers compared to the global model, the clients apply the cross-layer parameter sharing (Lan et al., 2020) to imitate the auxiliary models as the global model in the local training process. Each shared layer is concatenated with the corresponding tunable soft prompts. In this way, the tunable soft prompts in the server and clients are served in the same manner in an FL course.

During the local training process, clients adopt an alternative training method to achieve both *global model alignment* and *local knowledge capturing*, as shown in Figure 1 (b). To be more specific, firstly clients concatenate the received soft prompts with the local auxiliary models, and finetune the parameters of the auxiliary models while

Table 1: The comparisons between the proposed FEDSP and baselines with GPT2-XL.

| Methods | ARC-C | ARC-E | HellaSwag | OpenBookQA | PIQA | RACE | SciQ |
|---|---|---|---|---|---|---|---|
| ZERO-SHOT | 25.1 | 58.2 | 40.0 | 23.0 | 70.9 | 33.0 | 83.2 |
| FINETUNE | 30.0 | 62.9 | 40.7 | 30.0 | 73.2 | 43.2 | 92.5 |
| PREFIX-TUNING | 28.2 | 58.9 | 40.4 | 25.6 | 72.3 | 38.2 | 92.9 |
| FEDPROMPT | **27.5** | 59.4 | 40.7 | **26.2** | **71.9** | **38.3** | **92.8** |
| FEDPROMPT-SINGLE | 20.1 | 38.3 | 35.0 | 12.4 | 63.9 | 31.6 | 72.3 |
| FEDSP (ours) | 26.5 | **61.2** | **40.9** | 24.2 | 71.0 | 35.2 | **92.8** |
| *w/o* KD | 17.8 | 41.4 | 40.1 | 13.0 | 62.2 | 36.7 | 84.4 |
| *w/o* CS | 21.0 | 44.6 | 40.0 | 10.6 | 66.6 | 37.2 | 89.2 |
| *w/o* AT | 25.6 | 56.4 | 37.1 | 13.4 | 69.7 | 34.3 | 81.0 |

freezing the soft prompts. Since these soft prompts have been plugged into the global model and updated by the server before being shared, the purpose of freezing soft prompts and updating the auxiliary models is to align the representation of auxiliary models and the global model, with the help of the same soft prompts. Such an alignment is a necessary step in local training as the misaligned might cause a mismatch of the soft prompts between the server and clients.

After the global model alignment, clients perform prompt tunning as those used in PEFT algorithms, which implies that clients freeze the auxiliary models and only finetune the soft prompts for learning useful knowledge from local data, so-called local knowledge capturing. These updated soft prompts are sent back to the server, serving as messengers to deliver useful knowledge learned from participants' local data.

## 4 Experiments

### 4.1 Models and Datasets

We conduct a series of experiments with two popular large language models, i.e., GPT2-XL (Radford et al., 2019) and OPT-1.3B (Zhang et al., 2022). Specifically, both GPT2-XL and OPT-1.3B adopt the transformer-based architecture (Vaswani et al., 2017). GPT2-XL has 48 layers and 1.5 billion parameters, and OPT-1.3B has 24 layers and 1.3 billion parameters. For the benchmarking datasets, we adopt seven question-answering datasets for quantitative analysis, including ARC-C/E (Clark et al., 2018), HellaSwag (Zellers et al., 2019), OpenBookQA (Mihaylov et al., 2018), PIQA (Bisk et al., 2020), RACE (Lai et al., 2017), and SciQ (Welbl et al., 2017). We use accuracy as the evaluation metric in the experiments.

### 4.2 Baselines

We compare the proposed FEDSP with five different kinds of baseline methods, including:

- **ZERO-SHOT**, which directly evaluates the pre-trained language models without updating based on the downstream datasets.

- **FINETUNE**, which finetunes the entire LLMs on the downstream datasets, respectively.

- **PREFIX-TUNING** (Li and Liang, 2021), which adds tunable soft prompts and only finetunes them on the downstream datasets, respectively. Note that the parameters of LLMs are frozen during the finetuning process.

- **FEDPROMPT** (Zhao et al., 2022), which applies PREFIX-TUNING in the field of FL. Clients need to keep both the global model and the soft prompts and only update and exchange the soft prompts. Note that the parameters of the global model are frozen.

- **FEDPROMPT-SINGLE**, a variant of FEDPROMPT, which reduces the layer number of clients' global model to 1 to ensure the privacy protection of the global model. This setting is similar to that of FEDSP.

### 4.3 Implementation Details

The experiments are conducted on eight NVIDIA GeForce RTX 3090 GPUs. We implement the proposed approach and baseline methods based on Huggingface Transformers (Wolf et al., 2020) and FederatedScope (Xie et al., 2023), and conduct the evaluation with the support of lm-eval[2]. Following previous study (Zhao et al., 2022), we divide

---

[2]https://github.com/EleutherAI/lm-evaluation-harness

Table 2: The comparisons between the proposed FEDSP and baselines with OPT-1.3B.

| Methods | ARC-C | ARC-E | HellaSwag | OpenBookQA | PIQA | RACE | SciQ |
|---------|-------|-------|-----------|------------|------|------|------|
| ZERO-SHOT | 23.5 | 56.9 | 41.5 | 23.4 | 71.5 | 34.2 | 84.4 |
| FINETUNE | 27.7 | 61.3 | 42.7 | 31.4 | 75.2 | 37.0 | 92.5 |
| PREFIX-TUNING | 27.4 | 57.7 | 41.9 | 27.2 | 73.2 | 38.5 | 89.6 |
| FEDPROMPT | **26.8** | 57.6 | 41.8 | **27.4** | **72.9** | **38.4** | 89.8 |
| FEDPROMPT-SINGLE | 24.6 | 54.4 | 38.3 | 19.0 | 68.8 | 30.4 | 80.5 |
| FEDSP (ours) | **26.8** | **58.0** | **42.1** | 26.0 | 71.9 | 36.6 | **89.9** |
| *w/o* KD | 22.1 | 53.5 | 40.8 | 17.6 | 68.0 | 32.2 | 80.7 |
| *w/o* CS | 24.3 | 53.5 | 40.6 | 18.0 | 69.3 | 36.0 | 76.5 |
| *w/o* AT | 26.2 | 56.4 | 35.1 | 21.8 | 71.0 | 33.2 | 89.8 |

each dataset into 10 partitions, and each client owns one of them. A linear decay learning rate scheduler with a warm-up proportion of 0.1 is used in the experiments. We adopt AdamW (Loshchilov and Hutter, 2019) as the optimizer with $\beta_1 = 0.9$, $\beta_2 = 0.999$, the weight decay is set to 0.01, and the batch size is set to 16. The layer number of the auxiliary model used in FEDSP is set to 1.

We conduct the grid search for the optimal hyperparameters. Specifically, the learning rate is tuned in {1e-4, 2e-4, 5e-4}, the training round is tuned in {20, 50, 100, 200}, and the local training step is tuned in {10, 20}. For the methods that adopt prefix-tunning, we set the dimension of the tunable soft prompts to 40. And we conduct knowledge distillation for 5000 training steps with an initial learning rate of 5e-4. Inspired by previous study Li and Liang (2021), we adopt the reparametrization strategy with a hidden size of 512 on HellaSwag, RACE, and SciQ datasets.

### 4.4 Experimental Results

**Comparisons**  The comparison results between FEDSP and baseline methods are demonstrated in Table 1 and 2. From these results we can observe that, FINETUNE brings a large performance boost on the downstream tasks compared to those results achieved by ZERO-SHOT, and costs lots of computation resources. Therefore, PREFIX-TUNING, which only tunes soft prompts, achieve comparable performance but needs much fewer resource than FINETUNE. These results are consistent with previous studies (Li and Liang, 2021; He et al., 2021) and show the effectiveness of PEFT techniques.

When applying FL for privacy protection, we can see that FEDPROMPT achieves similar performances with PREFIX-TUNING on all the downstream datasets, demonstrating the effectiveness of

FL algorithms in maintaining the performance compared to the central training algorithms. However, there exists a noticeable gap between the performance achieved by FEDPROMPT and FINETUNE, which is attributed to the fact that the number of tunable parameters in FEDPROMPT is much fewer than those in FINETUNE.

Further, FEDPROMPT-SINGLE performs significantly worse than FEDPROMPT, and even worse than ZERO-SHOT on almost all the adopted datasets. These experimental results further confirm that the misalignment between the server's and clients' models might bring the optimization gap and make the federal training process meaningless, as discussed in Section 3.2.

The proposed FEDSP achieves much better performance compared to FEDPROMPT-SINGLE, and is competitive with those of FEDPROMPT and PREFIX-TUNING. For example, evaluated on the ARC-C dataset with GPT2-XL/OPT-1.3B, the performance of FEDSP is significantly better than FEDPROMPT-SINGLE with an improvement of 6.4%/2.2%, and is similar to FEDPROMPT within 1% performance gap. These results show the effectiveness and advantage of FEDSP in solving the misalignment issue and using tunable soft prompts as messengers for knowledge delivery. Meanwhile, with the help of these tunable soft prompts, it is worth pointing out that the proposed FEDSP provides both privacy protection for local data and global model, and needs fewer computation and communication costs compared to baselines.

**Ablation Study**  We conduct an ablation study to show the contributions of different techniques used in the proposed FEDSP. We separately remove the effect of using knowledge distillation (w/o KD), cross-layer sharing (w/o CS), and alternative training (w/o AT). The experimental results are reported

Table 3: Model performance w.r.t. different selected layers of the auxiliary models.

|  | ARC-C | ARC-E | HellaSwag | OpenBookQA | PIQA | RACE | SciQ |
|---|---|---|---|---|---|---|---|
| GPT2-XL$_{BOT}$ | **26.5** | **61.2** | **40.9** | **24.2** | **71.0** | 35.2 | **92.8** |
| GPT2-XL$_{MID}$ | 22.7 | 60.9 | 39.6 | 15.0 | 70.4 | 35.3 | 90.9 |
| GPT2-XL$_{TOP}$ | 25.2 | 59.4 | 39.2 | 9.6 | 70.8 | **35.6** | 86.3 |
| OPT-1.3B$_{BOT}$ | **26.8** | **58.0** | **42.1** | **26.0** | **71.9** | **36.6** | **89.9** |
| OPT-1.3B$_{MID}$ | 27.0 | 56.4 | 40.8 | 23.2 | 70.8 | 35.8 | 89.5 |
| OPT-1.3B$_{TOP}$ | 26.5 | 56.6 | 39.1 | 24.0 | 70.9 | 35.6 | 89.4 |

at the bottom of Table 1 and 2.

From these results, we can observe that removing any one of the adopted techniques brings a significant performance drop on most of the adopted datasets. For example, when removing KD/CS/AT on the ARC-C dataset, the model performance is decreased by 8.7%/5.5%/0.9% for GPT2-XL and 4.7%/2.5%/0.6% for OPT-1.3B, respectively. These results show that the techniques are all necessary in FEDSP and have positive effects for further improving overall model performance.

### 4.5 Further Discussions

In this section, we provide further discussions to better understand the proposed FEDSP.

**Auxiliary Model** The auxiliary models are first produced by the server according to the global model, and then updated by the clients based on their local data independently. In this part, we compare different mechanisms for producing the auxiliary model, including varying the selected layer from the global model and the layer numbers. The specific layer of bottom/middle/top is 1/24/48 and 1/12/24 for GPT2-XL and OPT-1.3B, respectively

As shown in Table 3, the server can select one of the layers from the bottom (denoted as BOT), the middle (denoted as MID), or the top (denoted as TOP) of the global model (i.e. LLMs) to produce the auxiliary model. We can observe from the table that the BOT selection strategy has the best performances on most datasets, and the other two strategies have closed performances. For example, compared with GPT2-XL$_{BOT}$/OPT-1.3B$_{BOT}$ on the ARC-E dataset, the performances of GPT2-XL$_{MID}$/OPT-1.3B$_{MID}$ and GPT2-XL$_{TOP}$/OPT-1.3B$_{TOP}$ have a degradation of 0.3%/1.6% and 1.8%/1.4%, respectively. These results imply the fact that the bottom layer is more suitable for achieving a good model alignment, as it is directly connected with the word embedding

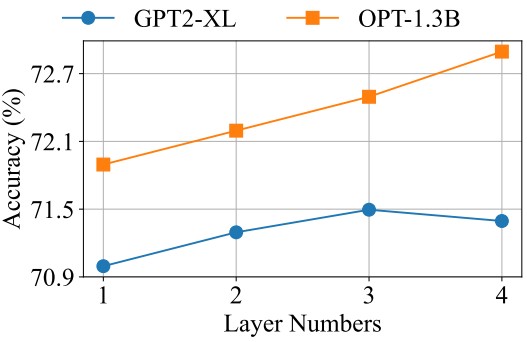

Figure 2: Model performance w.r.t. different layer numbers of the auxiliary models on PIQA dataset.

layer for learning low-level representations, and might change slighter than other layers during the finetuning process.

Besides, we vary the layer numbers of the auxiliary models, from 1 to 4, and report the experimental results in Figure 2. To apply the cross-layer sharing technique, the parameters of auxiliary models are shared $L/N$ times where $L$ is the total layer number of the global model and $N$ is the layer numbers of auxiliary models. We conduct evaluations on the PIQA dataset, which show the proposed FEDSP is able to achieve better performances with the increase of selected layer number. For example, based on GPT2-XL/OPT-1.3B, the performance of FEDSP on PIQA improves from 71.0/71.9 to 71.4/72.9 when the number of selected layers increases from 1 to 4. These results demonstrate that the proposed method is flexible to satisfy different requirements of computation resources from various applications, achieving a better balancing of privacy protection and model utility.

**Efficiency Comparisons** We compare the model sizes and communication costs of the proposed approach and baselines, as shown in Table 4 and 5. The model size refers to the number of model parameters loaded by the clients, and the commu-

Table 4: Efficiency comparisons between FEDSP and baselines on ARC-C with GPT2-XL.

| Method | Model Size | Comm. Cost |
|---|---|---|
| ZERO-SHOT | 1.6B | - |
| FINETUNE | 1.6B | 1.6B |
| FEDPROMPT | 1.6B | 6.1M (0.4%) |
| FEDSP | 111.1M (6.9%) | 7.2M (0.5%) |

Table 5: Efficiency comparisons between FEDSP and baselines on ARC-C with OPT-1.3B.

| Method | Model Size | Comm. Cost |
|---|---|---|
| ZERO-SHOT | 1.3B | - |
| FINETUNE | 1.3B | 1.3B |
| FEDPROMPT | 1.3B | 3.9M (0.3%) |
| FEDSP | 153.8M (11.8%) | 5.4M (0.4%) |

nication cost refers to the number of parameters being exchanged between the server and clients for each training round. Here to better show such comparisons, we assume that the communication cost of FINETUNE in the context of FL is the same as the model size of LLMs.

From the table, we can conclude the model size and communication cost of FEDSP are significantly less than FINETUNE (with degradation of 99.5%/99.6% and 93.1%/88.2% on GPT2-XL/OPT-1.3B, respectively). Compared with FEDPROMPT, FEDSP has similar communication cost while significantly reducing the model size loaded by the clients (with a decrease of 93.1%/88.2% on GPT2-XL/OPT-1.3B, respectively). These results demonstrate the efficiency of the proposed FEDSP for knowledge exchanging in an FL course, since clients only need to update the auxiliary model and soft prompts in the local training process.

**Privacy Protection** The proposed FEDSP provides fundamental privacy protection (i.e., no sharing data directly) with the help of the federated learning framework. As the number of the model parameters of soft prompts is much smaller than that of the entire model, we believe that the level of privacy-preserving of the proposed method would be better or at least at the same level. Further, privacy protection can be enhanced by privacy protection algorithms, such as differential privacy techniques (Triastcyn and Faltings, 2019; Wei et al., 2020), to satisfy flexible protection requirements from different applications.

## 5 Related Work

**Parameter-efficient Finetuning** With the increasingly larger scale of the pre-trained large language models (LLMs), finetuning the entire model becomes unaffordable for many individual researchers. As an efficient adaptation of LLMs to new downstream tasks, Parameter-efficient Finetuning (PEFT) (Houlsby et al., 2019; Lester et al., 2021; Li and Liang, 2021; Hu et al., 2021; Zaken et al., 2021; He et al., 2021; Khashabi et al., 2022) algorithms have emerged where only negligible proportions of parameters in the original LLMs are required to be updated. The main idea of PEFT is to add continuous soft prompts to the original model and only the parameters of the soft prompts need to be finetuned. For example, Liu et al. (2021) and Lester et al. (2021) propose to add a few continuous tunable parameters as soft prompts to the input word embeddings, and Li and Liang (2021) and Liu et al. (2022) suggest to prepend separate soft prompts to every model layer.

**Federated Learning in NLP** In order to protect data privacy, federated Learning (FL) has attracted a lot of attention from both academic and industrial (Konečný et al., 2016; McMahan et al., 2017; Yang et al., 2019; Kairouz et al., 2021). With more and more language assistance products being applied in real-world applications, FL has also increasingly appeared in the community of NLP to address the problem of privacy leakage, such as machine translation (Passban et al., 2022; Du et al., 2023) and question answering (Chen et al., 2021; Ait-Mlouk et al., 2022), and so on (Li et al., 2021; Lin et al., 2022; Kuang et al., 2023; Cai et al., 2023; Liu et al., 2023).

Besides data privacy, related works are also focused on the following challenges in the field of NLP: (i) Data heterogeneity. For example, Chen et al. (2021) proposes a federated matching framework with a backbone-patch architecture to address the non-identical and independent distribution (non-IID) problem of the training data. Li et al. (2022) propose to split the model into a local part and a global part to handle the non-IID issue. (ii) Task heterogeneity. For example, Dong et al. (2022a) propose an Assign-Then-Contrast framework to collaborate heterogeneous NLP tasks. Dong et al. (2022b) propose a few-shot FL framework with an energy-based weighting algorithm to enhance the cross-task generalization ability. (iii) Efficient

communication. For example, Passban et al. (2022) presents a dynamic pulling FL method to dynamically control the communication bandwidth. Du et al. (2023) presents a federated nearest neighbor framework to reduce the communication overhead.

Although FL can protect data privacy to some extent, it is incapable to cope with the situation when the server and clients need to protect their model privacy. In this paper, we make the first attempt to incorporate the idea of using tunable soft prompts as messengers to meet the requirement of protecting both data privacy and model privacy.

# 6 Conclusions

In this paper, we propose a novel FL training approach, named FEDSP, with tunable soft prompts as messengers to accomplish the knowledge delivery among participants. Since the proposed FEDSP does not need to share the global model, it provides the necessary privacy protection for the global model when finetuning LLMs. These soft prompts are broadcast to the clients at each training round, plugged into an auxiliary model, and used to capture the knowledge from local data. Extensive experiments on various benchmarking datasets demonstrate the effectiveness of FEDSP, showing that using tunable soft prompts as messengers in FL is able to protect both model privacy and data privacy, and achieve competitive performance with baselines and needs much less computation and communication costs.

## Limitations

We propose to use tunable soft prompts as messengers for useful knowledge exchange in FL. As for the limitations of this study, we conclude from the following three perspectives: (1) The proposed approach achieves privacy protection for both local data and the global model, but at the same time brings a slight performance drop compared to finetuning the LLMs directly. It can be regarded as a trade-off between privacy protection and model utility. We hope that further research can achieve a better balance on such a trade-off. (2) Some adopted techniques, such as cross-layer sharing, rely on the transformer-based architecture, which is the most widely used architecture in LLMs. In the long run, it could be better to choose a general technique to construct the auxiliary model used in FEDSP. (3) Although we conduct the first attempt to provide protection for both local data and global

models, a future direction can be how to further improve and adjust the privacy protection strength. For example, how to protect the model architecture of the global models, or how to utilize differential privacy techniques to satisfy various real-world applications with different protection requirements.

## Acknowledgements

This work was supported in part by the National Natural Science Foundation of China under Grant 61602013.

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
