# OpenReview forum: "Tunable Soft Prompts are Messengers in Federated Learning"
_EMNLP/2023/Conference — EMNLP 2023 Findings_

### Official Review · Reviewer_d21D · 2023-08-05

**Soundness:** 3

**Excitement:**

3: Ambivalent: It has merits (e.g., it reports state-of-the-art results, the idea is nice), but there are key weaknesses (e.g., it describes incremental work), and it can significantly benefit from another round of revision. However, I won't object to accepting it if my co-reviewers champion it.

**Missing References:**

The prompt finetuning in FL needs the following reference:

[1] Guo, Tao, et al. "PromptFL: Let Federated Participants Cooperatively Learn Prompts Instead of Models--Federated Learning in Age of Foundation Model." arXiv preprint arXiv:2208.11625 (2022).

[2] Zhao, Haodong, et al. "Reduce communication costs and preserve privacy: Prompt tuning method in federated learning." arXiv preprint arXiv:2208.12268 (2022).

**Paper Topic And Main Contributions:**

This paper introduces a topic called model privacy in this large language model (LLM) period, which is interesting and somehow necessary in academia. To protect model privacy in FL, the authors utilize knowledge distillation initialization, cross-layer parameter sharing, alternative training and parameter efficient fine-tuning. By testing on two datasets and two LLMs, the author illustrates a way of LLM training in FL with model privacy protected.

**Questions For The Authors:**

1. What is the meaning of the $auxiliary\ model$? What is the definition of this term? If this term is from a paper, a reference is needed. Or if it is created by the author, a definition or some examples at least should be given. Is the auxiliary model a part of LLM? Is the local model the same as the auxiliary model? Is the local model's structure the same as the global model? In the standard FL, the global model should be the same as the local model for weight aggregation and updating. For model privacy, the global model cannot share its weight with the clients. I am puzzled about whether the global model is the same as the local model.
2. Why is there centralized training in the baselines? If this paper is about FL, centralized training is very likely better than FL.
3. What is the model structure of the global model and the client model in your experiments?
4. If there is no knowledge distillation (KD), how can you initialize the local model since you want to protect the model privacy in Table 1 and Table 2?
5. If no alternative training (AT) exists, how can you train the local model in Table 1 and Table 2?
6. What is the specific layer of bottom, middle, and top in Table 3?

**Reasons To Accept:**

1. The idea itself is interesting. Model privacy is a problem in this period if we want to use FL but do not want to share the model weight.
2. The discussion is detailed.
3. The code is complete for reproducing the results.
4. The experiments are complete.

**Reasons To Reject:**

1. The writing is hard to follow.  There is no contribution list at the end of Introduction part. I read it several times but I am sorry that I cannot catch your theme. Is this paper mainly about model privacy in FL (FedSP) or soft prompt usage?
2. If the theme is mainly about FedSP, the performance of FedSP is not the best in Table 1 and Table 2 on some datasets.
3. If the paper is about FedSP, the author should do more experiments about FL settings, like the number of clients, communication rounds, etc.
4. The settings of the global and client model are not clear, like the model structure, etc.

**Reproducibility:**

5: Could easily reproduce the results.

**Reviewer Confidence:**

4: Quite sure. I tried to check the important points carefully. It's unlikely, though conceivable, that I missed something that should affect my ratings.

---

> ### Author Rebuttal · Authors · 2023-08-28
>
> Thank you very much for your detailed comments and helpful suggestions! We make the following responses point by point to address your comments. We use **R** to denote reasons to reject and **Q** to denote questions for the authors.
>
> **R1: "There is no contribution list at the end of Introduction part."**
>
> Thank you for the comments! The contributions of this paper are summarized as:
>
> - In this study, what we focus on is a feasible solution when the global model in federated learning cannot be shared due to some commercial or privacy concerns. Such a scenario is reasonable and practical, but could not be handled by previous studies.
> - The proposed method, called FedSP, is able to accomplish information exchange among participants via tunable soft prompts. These soft prompts are updated and transmitted between the server and clients, taking over the duty of the global model parameters and serving as messengers. FedSP makes it possible and meaningful to apply federal training when the global model cannot be shared (e.g., closed-source LLMs).
> - We conduct a series of experiments with two LLMs (GPT2-XL and OPT-1.3B), showing that FedSP achieves competitive performance compared to baseline methods. We believe that this work is an important forward based on previous studies to handle the practical issues in some applications that own secret models such as healthcare.
>
> We will add these contributions accordingly in the latest version for better understanding. Thank you again for your valuable suggestions to make our paper better!
>
>
> **R2: "The performance of FedSP is not the best in Table 1 and Table 2 on some datasets."**
>
> Thanks a lot for your comments!
>
> - The baseline methods include FINETUNE (centralized training entire model),  PREFIX-TUNING (centralized training soft prompts) and FEDPROMPT (training soft prompts with FedAvg). These methods actually have a more relaxed assumption on the privacy concerns (e.g., sharing the data or global model) compared to FedSP (data and global model could not be shared), so it is not surprising that FedSP is not the best in the reported results.
> - As we all know that there exists a trade-off between model utility and privacy protection, it is still important and meaningful to compare FedSP with these baselines to show how the proposed method balances such trade-off. The experiments show that FedSP achieves competitive performance compared to baseline methods, which implies that FedSP is an effective solution to satisfy a more strict privacy assumption with acceptable performance changes.
>
> We will add the above discussion in the latest version to make them more clear, thank you again!
>
>
> **R3: "The author should do more experiments about FL settings, like the number of clients, communication rounds."**
>
> Thanks a lot for your helpful suggestions! We will report more experimental results in the latest version, thanks again.
>
>
> **R4: "The settings of the global and client model are not clear, like the model structure."**
>
> Thanks a lot for your comments! The global model is the transformer-based language model (e.g., GPT2-XL or OPT-1.3B), which contains an embedding layer, several similar transformer blocks, and an output layer (please refer to section 4.1 or [1, 2] for more details). For the client model, it is similar to the global model but only contains one layer. Both the global model and client model are added with some soft prompts following [3], where continuous prompts are applied for every layer input of the language model.
>
> We will add these details to make them more clearly in the latest version, thank you again.
>
>
> **Q1: "What is the meaning of the auxiliary model? What is the definition of this term?"**
>
> Thanks a lot for your comments! The auxiliary model is used for knowledge communication, together with soft prompts, which is one corresponding layer of the global model. We use the "auxiliary model" to distinguish it from the global model (i.e., the entire model).
> We will add these details to make them more clearly in the latest version, thank you again!
>
>
> **Q2: "Why is there centralized training in the baselines?"**
>
> Thanks a lot for your comments!
>
> - We add centralized training in the baselines to show that FedSP achieves a good balance between privacy and utility. In other words, the centralized training methods can be regarded as the oracle to FL methods. Such comparisons are also used in previous studies.
> - For more details about the baselines, please refer to R2: "The performance of FedSP is not the best in Table 1 and Table 2 on some datasets.".
>
> We will add the above discussion in the latest version to make them more clear, thank you again!
>
>
> **Q3: "What is the model structure of the global model and the client model in your experiments?"**
>
> Thanks a lot for your comments! Please refer to R4: "The settings of the global and client model are not clear, like the model structure."
>
>
> **Q4: "If there is no knowledge distillation (KD), how can you initialize the local model since you want to protect the model privacy in Table 1 and Table 2?"**
>
> Thanks a lot for your comments! When there is no KD, the local model is initialized from the corresponding layers of the global model. As the layer number of the local model is much smaller than that of the entire model, and other layers would not be shared during the whole training process, we believe that it would not bring additional model privacy issues compared to previous studies that share the entire global  model.
>
> We will add the above discussion in the latest version, thank you again!
>
>
> **Q5: "If no alternative training (AT) exists, how can you train the local model in Table 1 and Table 2?"**
>
> Thanks a lot for your comments! When no AT exists, it implies that we only finetune the soft prompts in each local model while keeping other model parameters frozen. We will add these details to make them more clearly in the latest version, thank you again.
>
>
> **Q6: "What is the specific layer of bottom, middle, and top in Table 3?"**
>
> Thanks a lot for your comments! The specific layer of bottom/middle/top is 1/24/48 and 1/12/24 for the LLM of GPT2-XL and OPT-1.3B, respectively. We will add these details to make them more clearly in the latest version, thank you again.
>
> We believe this submission has been further improved according to your helpful suggestions, and wish the above responses can answer all your questions and address all your concerns! Hope these responses can convince you to lean more toward acceptance of the paper, thank you again for the detailed comments!
>
>
> Refs:
> [1] Radford et al. Language Models are Unsupervised Multitask Learners. 2019.
> [2] Zhang et al. OPT: Open Pre-trained Transformer Language Models. arXiv preprint arXiv:2205.01068.
> [3] Liu et al. P-Tuning v2: Prompt Tuning Can Be Comparable to Fine-tuning Universally Across Scales and Tasks. ACL 2022.

---

### Official Review · Reviewer_Mx16 · 2023-08-06

**Soundness:** 3

**Excitement:**

3: Ambivalent: It has merits (e.g., it reports state-of-the-art results, the idea is nice), but there are key weaknesses (e.g., it describes incremental work), and it can significantly benefit from another round of revision. However, I won't object to accepting it if my co-reviewers champion it.

**Paper Topic And Main Contributions:**

The paper proposes a novel federated learning (FL) approach, named FedSP, that uses tunable soft prompts as messengers to exchange knowledge among participants without sharing the global model. FedSP involves initializing an auxiliary model for each client, sending up-to-date soft prompts to selected clients, and performing an alternative local training process for global model alignment and local knowledge capturing. The paper conducts experiments with two LLMs (GPT2-XL and OPT-1.3B) on seven question-answering datasets and shows that the proposed approach achieves competitive performance with baselines while reducing model size and communication cost.

**Questions For The Authors:**

Please address the concern in the Reasons To Reject section.

**Reasons To Accept:**

1. Protecting the privacy of large models in FL is an interesting research problem worth studying.
2. The paper is well-written, and the methods are clearly presented.
3. The authors have validated the effectiveness of FedSP on seven datasets and promised to release the source code in the future.

**Reasons To Reject:**

The paper does not have many issues. The only concern is about the construction of the auxiliary model. The author used a part of the global model to construct the auxiliary model, which I suspect may also include the word embedding layer that already contains a significant amount of information from the original LLM. A more appropriate approach would be to use a smaller, open-source LLM that is completely unrelated to the global model for experimentation.

**Reproducibility:**

5: Could easily reproduce the results.

**Reviewer Confidence:**

4: Quite sure. I tried to check the important points carefully. It's unlikely, though conceivable, that I missed something that should affect my ratings.

---

> ### Author Rebuttal · Authors · 2023-08-28
>
> Thank you very much for your detailed comments and helpful suggestions! We make the following responses point by point to address your comments. We use **R** to denote reasons to reject and **Q** to denote questions for the authors.
>
> **R: "The only concern is about the construction of the auxiliary model."**
>
> Thank you for the comments.
>
> - Compared to other model parameters, the word embeddings have much fewer parameters. Thus we believe that the level of privacy-preserving of the proposed method would be better (or at least no worse than) previous studies that propose to share the whole model parameters or gradients.
> - It is a very nice idea to adopt other LLMs for better privacy preservation. In fact, there might exist a trade-off between model utility and model privacy here, since using the same LLM might achieve a better model performance as the relationships between server's model and client's model, while using a different LLM can be a better choice for privacy protection. We will add more discussion in the latest version and consider these experiments as future work.
>
> Thank you again for the comments! We believe this submission has been further improved according to your helpful suggestions, and hope that these responses can address all your concerns and convince you to lean more toward acceptance of the paper.

---

### Official Review · Reviewer_4Hpf · 2023-08-07

**Soundness:** 3

**Excitement:**

2: Mediocre: This paper makes marginal contributions (vs non-contemporaneous work), so I would rather not see it in the conference.

**Missing References:**

Cai, Dongqi, et al. "FedAdapter: Efficient Federated Learning for Modern NLP." arXiv preprint arXiv:2205.10162 (2022).
Liu, Yi, et al. "Communication Efficient Federated Learning for Multilingual Neural Machine Translation with Adapter." arXiv preprint arXiv:2305.12449 (2023).
Lin, Bill Yuchen, et al. "Fednlp: Benchmarking federated learning methods for natural language processing tasks." arXiv preprint arXiv:2104.08815 (2021).
Li, Qinbin, Bingsheng He, and Dawn Song. "Model-contrastive federated learning." Proceedings of the IEEE/CVF conference on computer vision and pattern recognition. 2021.
Yoon, Tehrim, et al. "Fedmix: Approximation of mixup under mean augmented federated learning." arXiv preprint arXiv:2107.00233 (2021).

**Paper Topic And Main Contributions:**

Federated learning (FL) enables multiple participants to collaboratively train machine learning models from decentralized data sources, alleviating privacy concerns raised by directly sharing local data. However, the lack of protection for model privacy in FL becomes an unneglectable challenge, especially when people want to federally finetune models based on a private large language model. This paper proposes a novel FL training approach that accomplishes information exchange among participants via tunable soft prompts. These soft prompts are updated and transmitted between the server and clients, taking over the duty of the global model parameters and serving as messengers to deliver useful knowledge in the local data and global model.

**Questions For The Authors:**

(1) Please discuss the relation between this work and

Cai, Dongqi, et al. "FedAdapter: Efficient Federated Learning for Modern NLP." arXiv preprint arXiv:2205.10162 (2022).
Liu, Yi, et al. "Communication Efficient Federated Learning for Multilingual Neural Machine Translation with Adapter." arXiv preprint arXiv:2305.12449 (2023).

(2) Is it possible the soft prompt in the developed federated learning framework may also leak some important information, when some approaches are used to recover the information from the soft prompts?

Khashabi, Daniel, et al. "Prompt Waywardness: The Curious Case of Discretized Interpretation of Continuous Prompts." Proceedings of the 2022 Conference of the North American Chapter of the Association for Computational Linguistics: Human Language Technologies. 2022.

(3) In terms of performance, how does the proposed approach perform, when being compared to FedAvg, FedProx and FedOPT mentioned in

Lin, Bill Yuchen, et al. "Fednlp: Benchmarking federated learning methods for natural language processing tasks." arXiv preprint arXiv:2104.08815 (2021).

**Reasons To Accept:**

(1) The paper studies an emerging and important problem.

(2) The proposed solution has good intuitions and is easy to understand.

(3) The paper is well written and presented.

**Reasons To Reject:**

(1) The baselines related to federated learning seems to be limited. Another type of works, where adapter-based approaches are also developed in the federated learning, can be discussed or compared. Some classic federated learning approaches can also be used for NLP tasks and compared to the federated learning framework proposed in the paper.

Cai, Dongqi, et al. "FedAdapter: Efficient Federated Learning for Modern NLP." arXiv preprint arXiv:2205.10162 (2022).
Liu, Yi, et al. "Communication Efficient Federated Learning for Multilingual Neural Machine Translation with Adapter." arXiv preprint arXiv:2305.12449 (2023).
Lin, Bill Yuchen, et al. "Fednlp: Benchmarking federated learning methods for natural language processing tasks." arXiv preprint arXiv:2104.08815 (2021).
Li, Qinbin, Bingsheng He, and Dawn Song. "Model-contrastive federated learning." Proceedings of the IEEE/CVF conference on computer vision and pattern recognition. 2021.
Yoon, Tehrim, et al. "Fedmix: Approximation of mixup under mean augmented federated learning." arXiv preprint arXiv:2107.00233 (2021).

(2) Although it is claimed in the paper that ‘FEDSP does not need to share the global model, it provides the necessary privacy protection for the global model when finetuning LLMs.’, it is still unclear whether it is safe to pass the soft prompt (in natural, they are also model parameters) information. There are some works which try to recover discrete tokens from soft prompt parameters. It is unclear whether the soft prompt in the developed federated learning framework may also leak some important information, when some approaches are used to recover the information from the soft prompts. It seems that limited discussion are included in the paper.

Khashabi, Daniel, et al. "Prompt Waywardness: The Curious Case of Discretized Interpretation of Continuous Prompts." Proceedings of the 2022 Conference of the North American Chapter of the Association for Computational Linguistics: Human Language Technologies. 2022.

**Reproducibility:**

4: Could mostly reproduce the results, but there may be some variation because of sample variance or minor variations in their interpretation of the protocol or method.

**Reviewer Confidence:**

3: Pretty sure, but there's a chance I missed something. Although I have a good feel for this area in general, I did not carefully check the paper's details, e.g., the math, experimental design, or novelty.

---

> ### Author Rebuttal · Authors · 2023-08-28
>
> Thank you very much for your detailed comments and helpful suggestions! We make the following responses point by point to address your comments. We use **R** to denote reasons to reject and **Q** to denote questions for the authors.
>
> **R1: "The baselines related to federated learning seem to be limited. Another type of work, where adapter-based approaches are also developed in federated learning, can be discussed or compared. Some classic federated learning approaches can also be used."**
> **& Q1: "Please discuss the relation between this work and several previous studies."**
>
> Thank you for the comments! We are afraid that there exists a misunderstanding on the proposed method, since the discussed scenarios in this paper are significantly different from previous studies. For more details:
>
> - The mentioned related works are proposed to improve the communication efficiency in FL via applying parameter-efficient finetune techniques (such as adapters). We believe that these works make remarkable progress when the backbone model (i.e., the models that are frozen and maintained in both server and clients) are not secret, e.g., some open-source LLMs. However, when the backbone model could not be shared in FL due to some commercial or privacy concerns, these related works could not be applied.
> - What we propose in this paper is the feasible solution when the backbone model cannot be shared. We use an auxiliary model and soft prompts for local training and knowledge communication.
> - Note that similar settings (i.e., secret backbone model) are also discussed in previous studies [1] in the context of transfer learning, and such settings are reasonable as there exist some closed-source LLMs. We believe that this work is an important forward based on previous studies to handle the practical issues in some applications that own secret models such as healthcare.
>
> We will add more details in the latest version to avoid such misunderstandings, and hope the above discussion can address your concerns. Thank you again!
>
>
> **R2: "It is still unclear whether it is safe to pass the soft prompt (in natural, they are also model parameters) information."**
> **& Q2: "Is it possible the soft prompt in the developed federated learning framework may also leak some important information, when some approaches are used to recover the information from the soft prompts?"**
>
> Thanks a lot for your comments!
>
> - The proposed method provides fundamental privacy protection (no sharing of data directly) with the help of the federated learning framework. Meanwhile, the number of the model parameters of soft prompts is much smaller than that of the entire model, thus we believe that the level of privacy-preserving of the proposed method would be better or at least at the same level.
> - Besides, the proposed method is more robust than previous studies, since most of the backbone model would be shared in the proposed method (only one layer is shared after preprocessing) while previous studies are applied to open-source backbones.
> - Meanwhile, privacy protection can be enhanced by privacy protection algorithms (such as differential privacy, homomorphic encryption, etc.) to satisfy flexible protection requirements from different applications. Users can balance utility-privacy-cost according.
> - These privacy protection algorithms can be regarded as plugins and are orthogonal to the proposed framework, thus in this submission, we pay more attention to describing how to use the soft prompts to share knowledge without sharing the backbones.
>
> We will add the above discussions in the latest revision accordingly. Hope the above discussion can address your concerns, thank you again!
>
>
> **Q3: "In terms of performance, how does the proposed approach perform, when being compared to FedAvg, FedProx and FedOPT."**
>
> Thanks a lot for your comments! We have compared the proposed method with some closed related settings such as PREFIX-TUNING (training soft prompts centralized) and FEDPROMPT (training soft prompts with FedAvg). We agree that it is a good idea to apply different federated optimization algorithms (such as FedProx and FedOPT) in both baseline and the proposed method (in the submission, the proposed method adopts FedAvg for a fair comparison) for comparison. As it is an orthogonal direction to what we focus on in this submission, we leave it to future work.
>
> We will add the above discussions in the latest revision according to your suggestions! Thank you again!
>
>
> **For "missing references"**
>
> We really appreciate your suggestions on these related works. We will add them accordingly, together with discussions to highlight the differences and advances of the proposed methods. Thank you very much for helping us to make this submission better.
>
> Thank you again for the comments! We believe this submission has been further improved according to your helpful suggestions, and hope that these responses can address all your concerns and convince you to lean more toward acceptance of the paper.
>
>
> Refs:
> [1] Xiao et al. Offsite-Tuning: Transfer Learning without Full Model. arXiv preprint arXiv:2302.04870.

---

### Meta-Review · Senior_Area_Chairs · 2023-09-30

**Recommendation:** 2

**Metareview:**

The reviewers appreciated that the paper tackles an interesting and important problem. The paper also extensively evaluates their proposed approach. However, there were several concerns about how the work compares to different baselines (some of which the authors rebut are not applicable, esp. when the backbone model is private).

This is definitely an interesting direction and I encourage the authors to revise and improve their draft—for instance, it could benefit from clearer and more detailed presentation (reviewers note that they find it hard to follow, and some of the contributions are not immediately clear). The paper could also discuss some related work that was missed in this iteration.

---

### Meta-Review · Area_Chair_x7eD · 2023-10-05

**Recommendation:** 2

**Metareview:**

Reviewers acknowledged the paper's engagement with a significant and intriguing problem. They also commended the comprehensive evaluation of the proposed approach. Nevertheless, concerns arose regarding the comparisons with various baselines, with the authors noting that some are not applicable, particularly when the backbone model is private.

This direction is undeniably captivating, and I encourage the authors to refine and enhance their manuscript. Improvements could include a more lucid and detailed presentation to aid comprehension, addressing reviewers' difficulties in following the content and clarifying certain contributions. Additionally, the paper should consider incorporating relevant missed related work.

---

### Decision · Program_Chairs · 2023-10-07

**Decision:**

Accept-Findings

**Comment:**

The reviewers appreciated that the paper tackles an interesting and important problem. The paper also extensively evaluates their proposed approach. However, there were several concerns about how the work compares to different baselines (some of which the authors rebut are not applicable, esp. when the backbone model is private).

This is definitely an interesting direction and I encourage the authors to revise and improve their draft—for instance, it could benefit from clearer and more detailed presentation (reviewers note that they find it hard to follow, and some of the contributions are not immediately clear). The paper could also discuss some related work that was missed in this iteration.|Reviewers acknowledged the paper's engagement with a significant and intriguing problem. They also commended the comprehensive evaluation of the proposed approach. Nevertheless, concerns arose regarding the comparisons with various baselines, with the authors noting that some are not applicable, particularly when the backbone model is private.

This direction is undeniably captivating, and I encourage the authors to refine and enhance their manuscript. Improvements could include a more lucid and detailed presentation to aid comprehension, addressing reviewers' difficulties in following the content and clarifying certain contributions. Additionally, the paper should consider incorporating relevant missed related work.